# Low energy multiple blue light-emitting diode light Irradiation promotes melanin synthesis and induces DNA damage in B16F10 melanoma cells

**Siqi Zhou, Ryusuke Yamada, Kazuichi Sakamoto** [ORCID] *

Faculty of Life and Environmental Sciences, University of Tsukuba, Tsukuba, Ibaraki, Japan

* sakamoto@biol.tsukuba.ac.jp

## Abstract

Visible light is present everywhere in our lives. Widespread use of computers and smart-phones has increased the daily time spent in front of screens. What effect does this visible light have on us? Recent studies have shown that short-wavelength blue light (400-450nm) irradiation, similar to UV, inhibits the cell proliferation and differentiation, induces the intra-cellular oxidative stress, promotes the cell apoptosis and causes some other negative effects. However, it's unusual that directly face to such short-wavelength and high-energy blue light in daily life. Therefore, the effects of blue light with longer wavelength (470nm), lower energy (1, 2 J/cm$^2$) and multiple times (simulated daily use) exposure on cells have been studied in this experiment. In our results, low energy density multiple blue light inhib-ited cell proliferation and metastatic capability with a weak phototoxicity. Blue light also pro-moted intracellular reactive oxygen species and caused DNA damage. Furthermore, the melanin synthesis was also promoted by low energy density multiple blue light exposure. Together, these results indicate that longer wavelength and low energy density blue light multiple exposure is still harmful to our cells. Furthermore, prolonged exposure to screens likely induces dull skin through induction of melanin synthesis. These results further men-tioned us should paid more attention to controlling the daily use of digital device.

**Data Availability Statement:** All relevant data are within the paper and its Supporting Information files.

**Funding:** The author(s) received no specific funding for this work.

## Introduction

With the development of technology, we have to be exposed to more light in our daily life. Especially the long time use of television, computers, and smart phones further extend the time and increase the frequency our exposure to light. A recent web-based survey indicated that approximately 93.8% of respondents reported that their digital device usage hours increased under the influence of COVID [1]. In particular, the student population saw an increase of 5.18 ± 2.89 hours per day of screen-facing time due to e-learning [2]. It is well known that ultraviolet (UV) light from the sun can cause damage to cells, and even DNA [3]. Therefore, it is important to investigate the effect of light from the LED screen (there is a peak of light intensity in the 400–500 nm range of the screen spectrum) on our bodies and cells.

**Competing interests:** The authors have declared that no competing interests exist.

Recent research showed that red light with wavelengths from 600 nm to 760 nm can affect a wide range of activities in cells and tissues [4]. For example, it has protective effects on human epidermal fibroblasts [5] and prevents UV-induced damage in a nude mouse model [6]. And red light exposure does not cause oxidative stress and DNA damage [7, 8]. Other studies showed that red light promotes the proliferation and differentiation of human or rat mesenchymal stem cells [9, 10]. However, research about blue light with a wavelength of 400 nm to 495 nm showed negative effects to cells. For example, 415–455 nm high energy blue light affects retina cells causing dry eye cataract and age-related macular degeneration [11]. Blue light also inhibits proliferation and differentiation and induces apoptosis in bone marrow-derived mesenchymal stem cells [12]. Short wavelength blue light is toxic to skin keratinocytes and even non-toxic waves inhibit proliferation [13] and induce oxidative stress in live skin [7]. Blue light also negatively affects nematode lifespan [14]. However, in most of these studies, shorter wavelength (400–450 nm) and higher energy blue light is used, which is unusual to simulate the situation in our daily life. Therefore, we wondered how longer wavelength (470 nm) lower energy density and multiple exposure of blue light affect the physiological function of cells.

Oxidative stress is associated with many cellular responses, one of which is DNA damage in cells [15]. UV exposure can cause single-strand or even double-strand DNA breaks [3], and UV exposure can also lead to an increase in intracellular oxidative stress [16]. In view of the fact that high-energy blue light can cause cell damage in previous studies, we believe that it is necessary to explore whether blue light with longer wavelength and lower energy density can lead to the increase of reactive oxygen species in cells and DNA damage. Many studies have shown that red light can promote the movement and proliferation of cells [9, 10, 17]. However, compared to the negative effects of blue light on cells in other studies, a recent study showed that 470nm blue light can also promote wound healing effect in rat model [17]. Therefore, we wanted to verify the effect of blue light with longer wavelength and lower energy density on the ability of cell migration.

Melanin is a natural pigment synthesized by melanocytes [18]. Skin exposure to sunlight or UV irradiation triggers melanocytes in the epidermis to synthesize melanin to prevent damage caused by UV irradiation [19]. Melanin synthesis is mainly regulated by the rate-limiting enzyme tyrosinase [20]. Tyrosinase and tyrosine related proteins (TRP-1, TRP-2) catalyze tyrosine to synthesize melanin in the melanosome of melanocytes [21]. Tyrosinase gene expression is regulated by microphthalmia-associated transcription factor (MITF) transcription factor [22]. Previously, short wave high energy blue light has been found able to promote the melanin synthesis in melanocytes through up-regulating tyrosinase and MITF [23, 24]. So, we are also interested that how the longer wavelength and lower energy density regulate the melanin synthesis.

In this study, we investigated how longer wavelength and low energy density blue light multipes exposure affect the physiological effect (phototoxicity, proliferation, migration, ROS generation, DNA damage and melanin synthesis) on B16F10 melanoma cells.

## Materials and methods

### Cell and cell culture

B16F10 melanoma cell lines were obtained from the RIKEN Institute, Physical and Chemical Research Cell Bank (Tsukuba, Japan). Roswell Park Memorial Institute (RPMI)-1640 medium was purchased from Sigma-Aldrich (St. Louis, MO, USA). B16F10 melanoma cells were cultured in RPMI-1640 medium supplemented with 10% fetal bovine serum (Hyclone, USA). After a primary culture from frozen stocks in 10-cm dishes, the cells were seeded at densities

of $1.0 \times 10^5$ cells/ well in 60 mm dish, $3.0 \times 10^3$ cells/well in 24-well-plates, or $1.0 \times 10^3$ cells/well in 96-well-plates. The cells were allowed to attach to the bottom of the plates for 24 h before subsequent experiments.

## Light irradiation

The blue LED light (peak at 470nm, DC24V, 1.13A, Max 27.2W) used in this experiment is a product of AITECSYSTEM CO., LTD. (Japan). Blue light irradiation was performed in a separate incubator using the same conditions as a normal incubator: 37˚C, 5% $CO_2$. Blue LEDs were irradiated from a distance of 5 cm from the dishes or plates. Blue light intensity (1,000 lux, 0.27 mW/cm$^2$) was measured by an energy meter through the clear cover of a dish for 0, 1, and 2 h (0, 1 and 2 J/cm$^2$). For the 3-day experiment, the cells were exposed to blue light for 0, 1, and 2 hours per day and then returned back to the incubator (dark). After the irradiation of day 3, the cells were returned to the dark incubator for 24 hours before subsequent assays. HBSS (Wako, Japan) without phenol red was used in place of the medium during blue light treatment to prevent stimulating effects on cells. After blue light irradiation, HBSS was replaced with fresh RPMI-1640 medium and put back to dark incubator. The non light group was changed to HBSS or refreshed to medium together with irradiation groups, but the dish was kept in dark incubator.

## MTT assay

Cells were cultured in 24-well-plates as a density of $3.0 \times 10^3$ cells/well and treated with blue light for 0, 1, and 2 h each day for a total of 3 days. The enzymatic activity was measured daily after the irradiation 24 hours. After irradiation, cells were allowed to rest in the dark for 24 h, then the medium was replaced with fresh medium containing 500 μg/ml MTT (3-[4,5-dimethylthiazol-2-yl]-2,5 diphenyl tetrazolium bromide) reagent, and the cells were incubated at 37˚C for another 4 h. Isopropanol with 0.04 M HCl solution was added and the plates were maintained at 25˚C for 5 min. The colored solution was transferred to a 96-well assay plate, and the absorbance was measured at 570 nm using a Tecan microplate reader (Kawasaki, Japan) (isopropanol with 0.04 M HCl solution was used as the blank).

## Trypan blue assay and proliferation assay

Cells were cultured in 24-well-plates as a density of $3.0 \times 10^3$ cells/well. The cell survival rate was measured 24 h after blue light irradiation. Cells were collected with trypsin, centrifuged at 800 x g, and the cell pellet was resuspended in 100 μL phosphate buffered saline (PBS). Ten μL of the suspension and 10 μL of 0.4% trypan blue (Sigma, USA) were mixed for one minute, and the number of live and total cells was counted using a cell counter (Logos Biosystems, Korea). The numbers of live cells was used as the cell number for the growth curve.

## Apoptosis detection

Cells were prepared in a Nunc black wall clear flat bottom 96 well plate (Thermo Fisher Scientific, USA) as a density of $1.0 \times 10^3$ cells/well. The Apoptosis/Necrosis assay kit (Abcam, England) was used to measure cells that were treated with blue light (0, 1 or 2 h each day) for 3 days. The green fluorescence (Ex/Em = 490/525 nm) for apotosis cells and the blue (Ex/Em = 405/450 nm) for alive cells were stained after the day 3 irradiation 24 hours. Photos of the cells were taken by a fluorescence microscope (Keyence, Japan).

## Wound healing

$1.0 \times 10^5$ cells were seeded into 6 cm dishes until confluent. The cells were scratched with a yellow tip in a cross-like pattern and blue light was immediately irradiated: the area of the cross immediately after scratching (0 h) was designated S1 (labeled by yellow line), and the area of the cross after 18 h was S2 (labeled by yellow line). The migration of cells was measured as S1 –S2.

## DNA damage

Cells were seeded at $3.0 \times 10^3$ cells/ml in chamber slides and irradiated with blue light for 0, 1 and 2 hours. Immediately after irradiation, DNA damage was measured using the DNA damage detection kit—γH2AX–green (DOJINDO, Japan). Images were taken with a fluorescence microscope (Keyence, Japan) at Ex/Em = 494/518 nm. The fluorescence level was analyzed by Image J.

## ROS generation

Cells were prepared in a Nunc black wall clear flat bottom 96 well plate (Thermo Fisher, USA) as a density of $1.0 \times 10^3$ cells/well. Intracellular reactive oxygen species (ROS) were immediately measured after the irradiation with a DCFDA/H2DCFDA Cellular ROS assay kit (Abcam, England). Fluorescent photos were taken with a Keyence fluorescence microscope (Japan) at Ex/Em = 485/535 nm. The fluorescent signals were also measured using a BioTek fluorescence microplate reader (Tokyo, Japan).

## Melanin content assay

Cells for melanin content were prepared in a 6-well-plate as a density of $5.0 \times 10^4$ cells/well and treated with blue light for 3 days. After day 3 irradiation, cells were returned to the dark incubator for 24 hours and then collected with trypsin, centrifuged at 800 x g, and the cell pellet was dissolved in 1M NaOH. The sample was incubated at 80˚C for 1 h in a heat block to dissolve melanin. The absorbance of the melanin solution was measured using a BioTek microplate reader. The results were normalized according to the total protein levels measured using a Pierce™ BCA protein assay kit (Thermo Fisher Scientific, USA).

## Real-time PCR

Cells were prepared in 6 cm dishes as a density of $1.0 \times 10^5$ cells/dish and mRNA (immediately and 3 hours after blue light irradiation) was extracted using RNA iso Plus (Takara Bio, Japan). PrimeScript™ RT reagent kit with gDNA Eraser (Takara Bio, Japan) was used for reverse-transcription to cDNA. Quantitative PCR was performed with the THUNDERBIRD® SYBR® qPCR mix (Toyobo, Japan) and the assays were performed using Thermal Cycler Dice® Real Time System Lite (Takara Bio).

The amplification of target mRNAs was performed using specific primers (5′–3′): *mitf* (forward: GTGAGATCCAGAGTTGTCGT, reverse: AGTACAGGAGCTGGAG ATG); *tyrosinase* (forward: TGACTCTTGGAGGTAGCTGT, reverse: AACAA TGTCCCAAGTACAGG); *trp-1* (forward: AATGACAAATTGAGGGTGAG, reverse: GGCCTCTGAGGTTCTTTAAT); *trp-2* (forward: AGGAGTGAGGCCAAGTT ATGA, reverse: ATGAGAAACTGCCAACCTTA). *gapdh* (forward: TGCCGTTGAA TTTGCCGTGAGT, reverse: TGGTGAAGGTCGGTGTGAACGG) was used as internal reference for normalization.

### Statistical analysis

All experiments were repeated at least three times. The results are expressed as the mean ± SD. ANOVA was performed post hoc to compare data between the groups using SPSS Statistics. Statistical significance was set at a p-value of < 0.05.

## Results

### Low-energy density multiple blue light irradiations inhibited cell proliferation without causing apoptosis

MTT assay showed that longer wavelength and low energy density blue light irradiation significantly reduced cell enzymatic activity (Fig 1A), even only once irradiation the result of MTT was also decreased. Because MTT assay's result depends on the cell number and also the enzymatic acitivity, we also used trypan blue and cells proliferation to further investigate the phototoxicity of the blue light. Interestingly, cell survival rate did not significantly change due to blue light irradiation in the trypan blue assay (Fig 1B). While, blue light irradiation significantly reduced cell proliferation (Fig 1C). These results gave us an idea that low energy density blue light didn't kill the cells directly but inhibited the cell proliferation. Further, to certify the phototoxicity of blue light, we also measured the cell apoptosis. The number of apoptotic cells did not significantly increase indicating that blue light does not lead to apoptosis (Fig 1D and 1E).

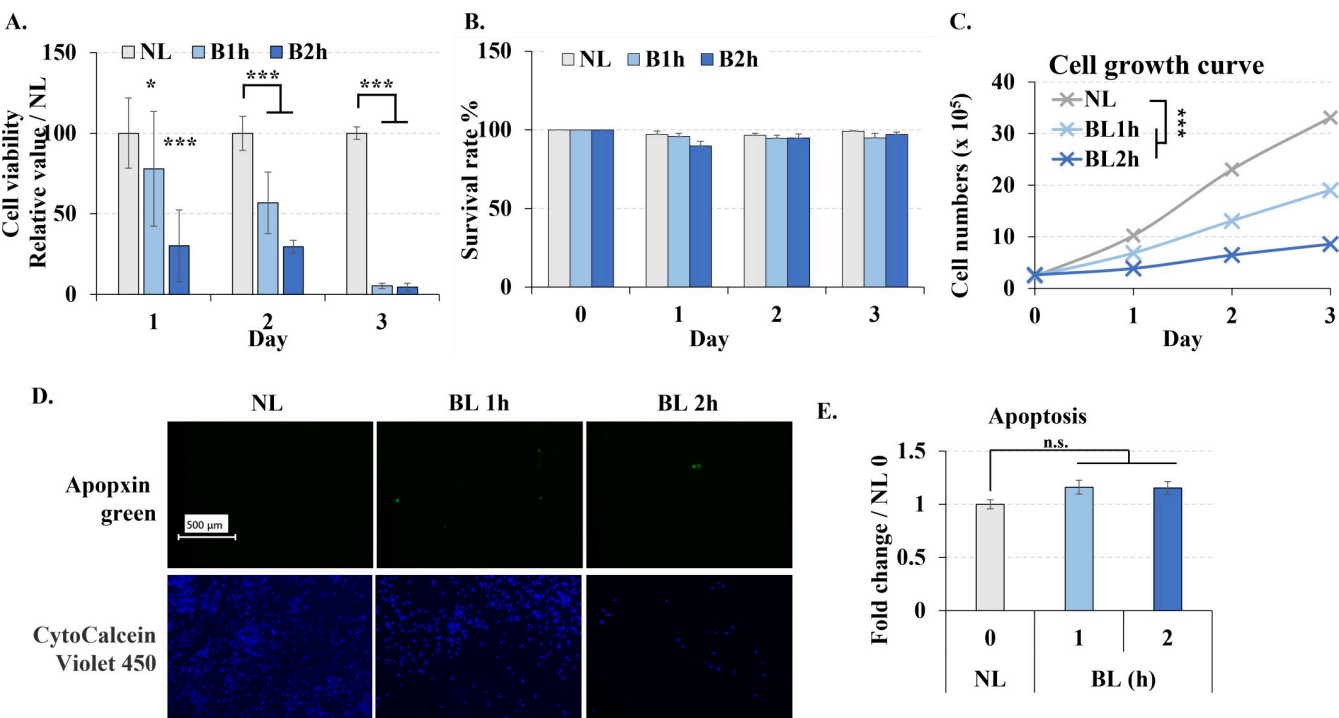

**Fig 1. Blue light effect on the viability of B16F10 melanoma cells.** Cells were treated with blue light (0, 1, and 2 h / day) for 3 days. **A.** Cell viability measured by MTT assays each day. **B.** Cell survival rate measured by trypan blue assays each day. **C.** Cell were irradiated once, returned to the dark incubator, and cell numbers were counted by cell counter every following day. **D.** Cells were immediately stained with apopxin green (apoptosis) and CytoCalcein Violet 450 (live cell) after the last irradiation, and images were taken with a fluorescent microscope. Scale bar, 500 μm. **E.** Fluorescence values for green were analyzed by image J. NL: non-light treatment; BL: blue light irradiation; B1h, B2h: blue light irradiation 1, 2 hours. The results are presented as the mean ± SD; n ≥ 3; *p < 0.05, ***p < 0.001 vs. NL 0.

## Low-energy density blue light inhibited cell migration

B16F10 mouse melanocytes have the ability to metastasize. In our study, the wound healing assay was used to assess the effects of blue light on melanoma cell migration *in vitro*. The yellow line was used to label the cross area without cells. We found that the area changes of the cross area without cells within 18 hours significantly decreased due to the blue light 1- and 2-hours irradiation (Fig 2), which means that longer wavelength low energy density blue light exposure inhibited the B16F10 melanoma cell migration.

## Low-energy density blue light induced ROS generation and DNA damage

UV irradiation can lead to the rise of ROS inside cells, therefore, we wondered whether longer wavelength low energy density blue light, which wavelength close to UV, also has an similar

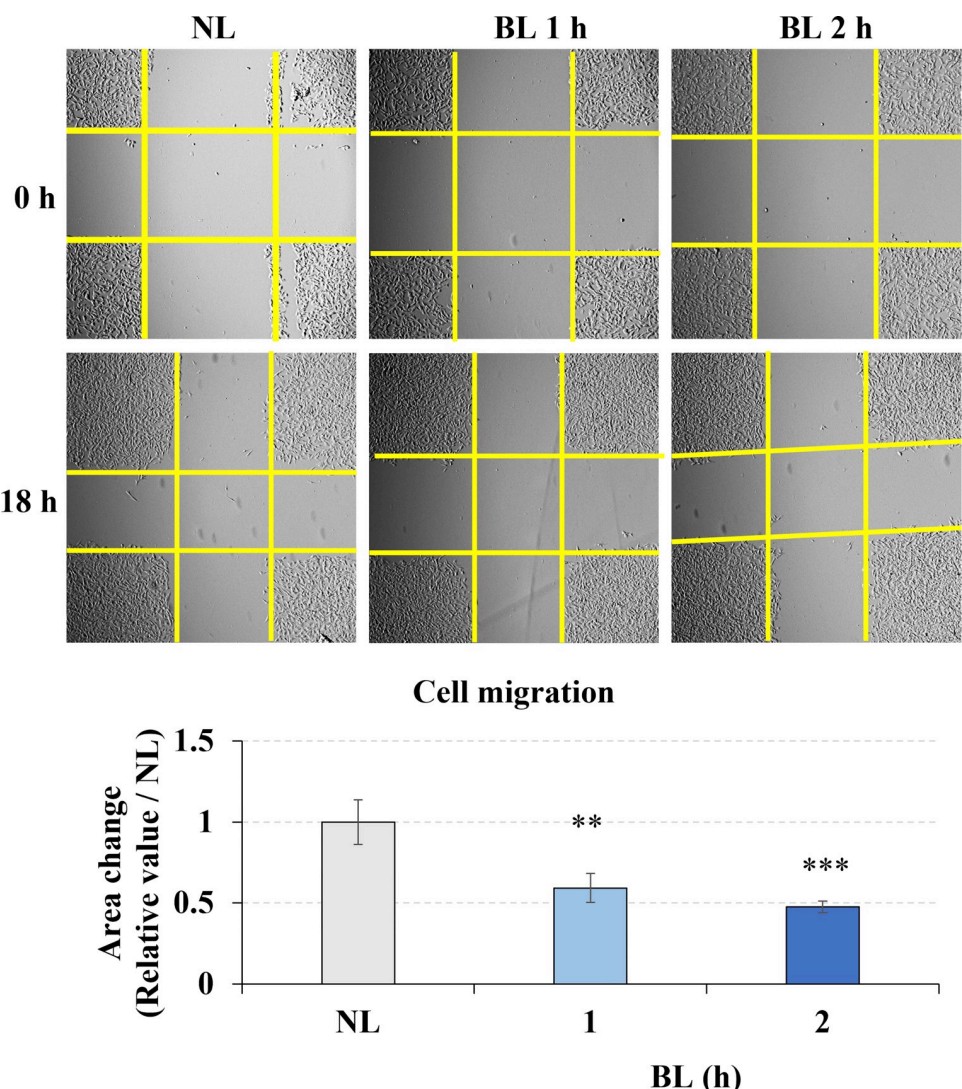

**Fig 2. Blue light effect on the migration of B16F10 melanoma cells.** Cells were irradiated with blue light for 0, 1, or 2 h after preparing the gaps using pipet tips. Yellow lines labeled the area without cells. The images at 0 h and 18 h were taken and the area change of cross gap was analyzed by image J. The bar graphs show the chage of the area between 0 h and 18 h. NL: non-light treatment, BL: blue light irradiation. The results are presented as the mean ± SD; n ≥ 3; **p < 0.01, ***p < 0.001 vs. NL 0.

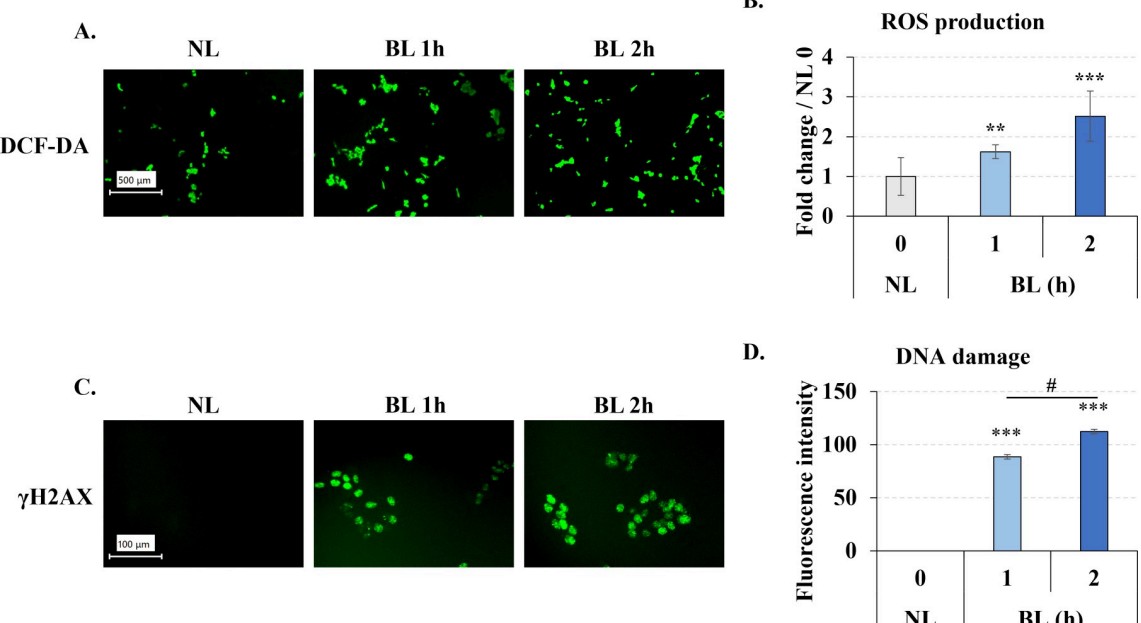

**Fig 3. Blue light induced ROS generation and DNA damage in B16F10 melanoma cells.** After irradiation, the cells were immediately assayed using the DCF-DA (**A, B**) or DNA damage kit (**C,D**), the pictures were taken by fluorescent microscope. The fluorescent of DCF-DA was measured by microplate reader (B). The images of DNA damage were analyzed by Image J. **A, B**. Blue light induced ROS generation in B16F10 melanoma cells. **C, D.** Blue light induced DNA damage in B16F10 melanoma cells. The length of the scale bar in **A** and **C** is 100 μm and 500 μm, respectively. Clear field of the cell after blue light irradiation was shown at S3 & S4 Figs in S1 File. NL: non-light treatment, BL: blue light irradiation. The results in B and D are presented as the mean ± SD; n ≥ 3; **p < 0.01, ***p < 0.001 vs. NL 0.

effect on intracellular ROS. After blue light 0, 1, 2h irradiation, we immediately measured the intracellular ROS by DCFDA staining. It is clear that the green fluorescence was increased in the blue light exposure groups (Fig 3A), the results by microplate reader also accordance with the microscope that blue light significantly increased the fluorescent in cells (Fig 3B). These results suggested that low energy density blue light, similar to UV light, increased intracellular ROS after irradiation.

Next we also measured the DNA damage, because ROS generated after UV irradiation of cells can damage DNA. Therefore, we also wanted to investigated the effect of blue light on DNA. After same exposure time with ROS generation assay, we collected the cells and measured the DNA damage with the "DNA Damage Detection Kit—γH2AX—GREEN" kit. Phosphorylated H2AX (γH2AX) is a marker of DNA damage. In this experiment, γH2AX is labeled and show a green fluoresence. We found the fluorescence of γH2AX was significantly increasd by the blue light exposure (Fig 3C and 3D), which suggested that low energy density blue light also caused intracellular DNA damage, which was increased with irradiation time.

## Low-energy density multiple blue light irradiation promoted melanin synthesis through tyrosinase up-regulation

In prior studies, short-wavelength(410–440 nm), high-energy (>20 J/cm$^2$) blue light has been shown to promote melanin synthesis. So we wanted to know if longer wavelength (470 nm), lower energy (1–2 J/cm$^2$), multiple (3 times) exposures to blue light would have the same result. In our results, low energy density multiple irradiations also slightly (about 1.2–1.3 times) but significantly promoted melanin synthesis (Fig 4A).

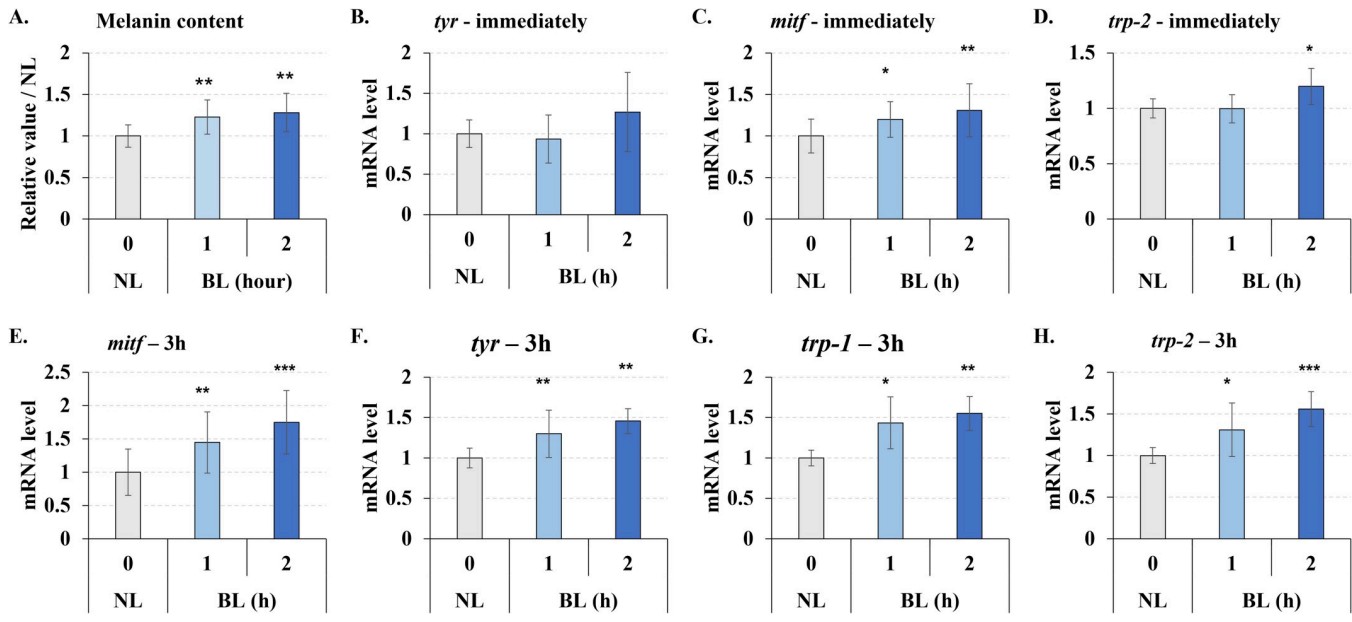

**Fig 4. Blue light enhanced melanogenesis in B16F10 melanoma cells.** Cells were treated with blue light (0, 1, and 2 h / day) for 3 days, and the cells from day 3 were collected. **A.** The melanin content increased by blue light irradiation. **B-D.** mRNA quantity determined immediately after irradiation (0 h) by real-time PCR. **E-H.** mRNA quantity 3 h after irradiation by real-time PCR. NL: non-light treatment, BL: blue light irradiation, tyr: tyrosinase. The results are presented as the mean ± SD; n ≥ 3; *p < 0.05, **p < 0.01, ***p < 0.001 vs. NL 0.

Furthermore, we examined the gene expression of factors that regulate melanin synthesis. There was no significant increase in the expression of genes related to melanin synthesis from cells that were immediately recovered after blue light exposure (Fig 4B & 4D), except for transcription factor *mitf* (Fig 4C). However, after blue light exposure 3 hours, *TYR*, *TRP-1*, *TRP-2*, and *mitf* expression were significantly up-regulated (Fig 4E–4H). Therefore, we concluded that low energy density multiple blue light exposure also increased melanin synthesis.

## Discussion

This study evaluated the ability of blue LED visible light to regulate the physiological effects of melanin synthesizing cells. Blue light can deeply penetrate into the skin layers and affect cells since it has a longer wavelength and lower energy compared to UV light [25]. Therefore, we need to start paying attention to the impact of daily lighting on ourselves, including the light found in mobile phones or computer screens.

Previous studies suggested that exposure to blue visible light showed negative effects to cells. Blue light is reported inhibits cell proliferation and increases ROS generation in bone marrow stem cells (BMSCs) [12], fibroblasts [26], HaCaTs (human keratinocytes) [27], retinal pigment epithelium cells [28], and U937 cells [29]. However, it is difficult to summarize the effect of blue light on inducing apoptosis from these studies.

Different wavelengths of blue light were used in previous studies. Shorter wavelengths of blue light such as 412–426 nm showed stronger toxicity [13, 26, 28], while 453 nm and 470 nm blue light had lower toxicity and inhibitory effects on cell proliferation and function [12, 13, 29]. Most experiments involving cells only involved one application of blue light [9–12, 25–29] which enabled a better understand of the regulatory effect of blue light on cells. However, it does not simulate real-world conditions using computers or mobile phones. Therefore, we used longer wavelength (470 nm) lower energy (1–2 J/cm$^2$) blue light (approximately the lux

of mobile phone screens and computer screens) and performed multiple exposures in this study.

Initially, we found the blue light irradiation decreased the cell enzymatic activity by MTT assay but didn't affect the cell survival rate by trypan blue assay. Since the MTT results were based on both cell number and enzymatic activity, which a decreased cell number was also found. It was difficult to determine whether the decrease in MTT was due to a decrease in cell number or a decrease in cell enzymatic activity. Therefore, we further performed trypan blue and cell proliferation assays. Intermittent, multiple irradiations didn't affect the cell survival rate but significantly reduced cell proliferation. These results indicated that the decreased enzymatic activity by blue light due to the reduced cell proliferation rather than cytotoxicity. We also detected the cyclin level after light exposure (S1 Fig in S1 File), the decreased cyclin mRNA also supports that the cell cycle was arrested by blue light. However, multiple low power irradiation did not increase apoptosis compared with other single-shot high-power irradiations at the same wavelength of 470 nm [12]. In order to further confirm that blue light didn't cause apoptosis, The expression of caspase-3 and the content of cleaved-caspase-3 were also detected (S2 Fig in S1 File). Although the expression level of caspase-3 increased, there was no band at the position of cleaved-caspase-3, which indicated that apoptosis did not occur. Longer wavelength lower energy blue light irradiation inhibited cell proliferation without inducing apoptosis, which is an interesting phenomenon. To cancer treatment, it means that it is possible to inhibit cancer growth without killing the cells. While further experiments are still necessary to verify this conjecture.

We found that a single irradiation of low-energy density blue light increased ROS levels in melanoma cells. UV light usually promotes intracellular ROS generation and induces DNA damage [3, 16]. The rise of ROS is often linked to DNA damage [15]. DNA damage promotes ROS generation via the H2A.X-Nox1/Rac1 pathway [30]. In our results, both ROS generation and DNA damage were up-regulated by blue light exposure, which suggested that although low energy density blue light didn't induce apoptosis, it also induce the oxidative stress and DNA damage. ROS regulates many intracellular activities, while excess ROS induces intracellular damage, including DNA damage [15]. DNA damage caused by ROS is divided into single-strand break (SSB) and double-strand break (DSB) [31]. DNA damage also causes cell cycle arrest and DNA repair [15]. Severe double-strand break of DNA damage directly leads to cell apoptosis [32]. These results help us understand why blue light causes DNA damage and reduces cell proliferation without causing apoptosis. We think the reason is the relatively low energy density blue light used in our experiments, the energy isn't strong enough to cause fatal damage to the cells. These results also remind us that we need to pay more attention to the daily use of digital device. Although there will be no serious harm, continuous long-term exposure will also lead to an increase in oxidative stress and a decrease in the proliferation of skin cells. Since we detected ROS and DNA damage at the same time point after light exposure in this experiment, it is still difficult to determine whether the DNA damage is directly caused by irradiation or by the ROS generated after light exposure. This point needs to be verified by further experiments in the future.

It is reported that ROS promotes cell migration [33]; however, our results found that blue light inhibited cell migration, which correlated with a previous study [26]. ROS also play an important role in cells, appropriate increased of hydrogen peroxide ($H_2O_2$) will promote intracellular signal conduction and thus promote cell migration [33, 34]. However, excessive ROS production will produce corresponding toxicity [35]. Thus, we think the type of ROS up-regulated by blue light was not $H_2O_2$, and the increase level was also higher than the necessary. For this reason, the rise of ROS and the DNA damage by blue light irradiation, caused the cell cycle arrest, furtherly negatively regulated the cell migration ability. However, it is still

necessary to identify the type of ROS production induced by blue light irradiation ($H_2O_2$ promotes cell migration). Therefore, further studies are required to investigate how blue light inhibits cell migration and determine the relationship between blue light induced ROS generation and inhibited migration.

Blue light promotes melanin synthesis at 450 nm and 60 $J/cm^2$ [23], or 415 nm and 50 $J/cm^2$ [24]. Our experiments verified the rise in melanin levels under longer wavelength low-energy density multiple irradiations of blue light (470 nm, 1 or 2 $J/cm^2$, 3 applications). *mitf* mRNA immediately increased following blue light irradiation, while the melanin synthesis-related factors *tyrosinase*, *trp-1*, and *trp-2* gradually increased and began to show 3 h after irradiation. We also detected the protein levels of tyrosinase, MITF, TRP-1 and TRP-2. Unfortunately, although we tested the samples after 0, 3, 12, 24, and three times of irradiation, it was still unable to get a stable change of the protein content. We suspect that this is due to disruption of cellular protein synthesis function due to DNA damage. Thus, it is necessary to conduct more research to investigate the mechanism and the target of blue light regulating melanin synthesis.

Intracellular ROS and DNA damage immediately occurred after blue light irradiation, and there was no significant change after three days irradiation in our previous study. This proved that low-energy density blue light is harmful to cells. The main function of melanin is to absorb UV, thereby inhibiting the intracellular damage caused by UV stimulation [36–38]. Intracellular melanin significantly increased after 3 d of blue light exposure. Therefore, we hypothesized that the ROS levels and DNA damage did not significantly change after 3 d due to the rise in melanin. However, further experiments are required to determine whether melanin has a protective effect against blue light-induced damage.

Last but not least, different wavelength of visible light has different penetration to the skin, and there is also a big difference between cells in a plate and the body. Also, the wound healing assay on rat previously suggested that both red and blue light irradiation promoted the wound healing [18], which was different with our results on migration. In order to investigate the function of blue light to the live skin and verify the depth with impact of blue light, more rigorous *in vivo*. experiments are still required.

## Conclusions

Multiple exposure to longer wavelength (470nm) low energy density (1 or 2 $J/m^2$) blue light didn't induce cell apoptosis but still inhibited the cell proliferation. This effect was induced by the increased ROS generation and DNA damage. But it is still necessary to investigate the mechanism how blue light induced the DNA damage and ROS generation. Longer wavelength low energy density multiple blue light exposure can also promote melanin synthesis. These results remind us that even low energy density blue light has a weak phototoxicity, repeated exposure will also cause cellular damage and regulate cellular function. It is better to pay more attention to the daily using time and frequency of the digital device to protect us with the damage from blue light.

## Supporting information

**S1 Checklist.**
(DOCX)

**S1 Graphic abstract.**
(TIF)

**S1 File.**
(PDF)

**S1 Raw images.**
(ZIP)

## Author Contributions

**Conceptualization:** Ryusuke Yamada, Kazuichi Sakamoto.

**Data curation:** Siqi Zhou, Kazuichi Sakamoto.

**Formal analysis:** Siqi Zhou, Kazuichi Sakamoto.

**Funding acquisition:** Kazuichi Sakamoto.

**Investigation:** Siqi Zhou, Ryusuke Yamada, Kazuichi Sakamoto.

**Methodology:** Siqi Zhou, Ryusuke Yamada.

**Project administration:** Kazuichi Sakamoto.

**Resources:** Siqi Zhou, Kazuichi Sakamoto.

**Supervision:** Kazuichi Sakamoto.

**Validation:** Siqi Zhou, Kazuichi Sakamoto.

**Visualization:** Kazuichi Sakamoto.

**Writing – original draft:** Siqi Zhou.

**Writing – review & editing:** Siqi Zhou, Kazuichi Sakamoto.

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
