## [Decision Letter · Decision Letter 0]

9 Nov 2022

PONE-D-22-27769Low Energy Multiple Blue light-emitting diode light Irradiation promotes melanin synthesis and induces DNA damage in B16F10 melanoma cellsPLOS ONE

Dear Dr. Sakamoto,

Thank you for submitting your manuscript to PLOS ONE. After careful consideration, we feel that it has merit but does not fully meet PLOS ONE’s publication criteria as it currently stands. Therefore, we invite you to submit a revised version of the manuscript that addresses the points raised during the review process.

We look forward to receiving your revised manuscript.

Kind regards,

Yi Cao

Academic Editor

PLOS ONE

Journal Requirements:

"No

"This work was supported in part by Grants-in-Aid for Scientific Research and Education from the University of Tsukuba, Japan."

"No

Reviewers' comments:

Reviewer's Responses to Questions

**Comments to the Author**

1. Is the manuscript technically sound, and do the data support the conclusions?

Reviewer #1: Yes

Reviewer #2: Partly

Reviewer #3: Partly

2. Has the statistical analysis been performed appropriately and rigorously? 

Reviewer #1: Yes

Reviewer #2: Yes

Reviewer #3: Yes

3. Have the authors made all data underlying the findings in their manuscript fully available?

Reviewer #1: Yes

Reviewer #2: Yes

Reviewer #3: Yes

4. Is the manuscript presented in an intelligible fashion and written in standard English?

Reviewer #1: Yes

Reviewer #2: Yes

Reviewer #3: Yes

5. Review Comments to the Author

Reviewer #1: Manuscript is well presented. This study provides important information that helps better understand the effect blue light has on cells. The abstract is clear and describes the background, goals, and results of the paper well. As a general comment, the grammar could be improved throughout the manuscript before the final submission. My specific comments are below:

INTRODUCTION

1) It may be beneficial to more clearly state the goals of this study. The last sentence of the introduction (page 4 line 51) states that “this study investigated the physiological effects of LED blue light on melanin synthesis in cells.” However, the paper examines many more things than just melanin synthesis. It also may be useful to provide more background in the introduction that will be later discussed in the paper, such as on wound healing, ROS, and DNA damage.

2) Page 3 Line 29 – The first sentence of the paper is confusing/ rather long, I would recommend to divide it into two sentences.

3) Page 3 line 39 – How is blue light significantly harmful to cells? Please provide more detail.

METHODS

1) MTT is an enzyme assay and not a viability assay, despite it often being mistakenly used as a viability assay. This may account for the discrepancy seen when comparing with trypan blue. Either the authors need to label the MTT as an enzyme function assay or remove from the manuscript.

2) Page 5 line 75 – Please be consistent with the use of “day” and “d” throughout.

3) Page 6 line 96 – How many hours each day were the cells treated with blue light?

4) Page 6 line 101 – Please change “confluence” to “confluent.”

RESULTS

1) Perhaps repeat the experiments in an additional cell line to ensure that these findings are not due to single cell line variability and are conserved across cell lines.

2) If blue light did not induce apoptosis, why was proliferation limited? The authors should add cell cycle experiments to determine blue light's effect on cell cycle. Additionally, is it possible that the cells were necrotic or fragmented and not picked up on flow cytometry?

3) Throughout the discussion and in the figure descriptions, authors did not say if findings were significant for numerous results (example – page 9 line 169, wound healing assay). Please always state if results are statistically significant.

4) How was the DNA damage assay selected? 

5) Page 8 line 151 – Authors state “longer blue light irradiation times reduced cell viability.” Please clarify how exactly long cells were irradiated in this result. And was this reduction in cell viability statistically significant?

6) Page 9 line 159 (Fig 1C) – Clarify which assay was used in the figure description.

7) Page 10 line 181– The first sentence in the ROS paragraph might fit better in the introduction.

8) Page 10 line 185 – Was intracellular ROS significantly increased?

9) Page 10 line 190 – State what kit was used to measure DNA damage.

10) Page 11 line 206 – Again, the first couple sentences of this paragraph belongs in the introduction, as this is the background/ reasoning for conducting these experiments

11) Page 11 line 209 – Authors state, “irradiation also slightly promoted melanin synthesis.” What does “slightly” mean?

DISCUSSION/ CONCLUSION

I believe this section could discuss the results of the experiments more. What are some possible reasons why low energy blue light inhibits migration? What is the significance of low energy blue light causing increased ROS and DNA damage to our cells? What are some potential reasons the author’s study found that blue light inhibits cell migration, when it is reported in other studies that ROS promotes cell migration?

How do the results of this study compare to other studies? The work here is very nice, and similar to the works referenced in manuscript reference #20 by Mamalis et al. Please note how you distinguished your works.

The authors also need to add more to the limitations section and how they mitigated against these challenges.

Additionally, the conclusion of the paper mostly restates the results. Are there any other final thoughts or conclusions that the authors would like readers to take away from this paper?

FIGURES

Figure 1C – This graph does not show if the results are statistically significant.

Reviewer #2: In this paper, low energy and multiple blue light irradiations were used to promote melanin synthesis and DNA damage in mouse melanoma B16-F10 cells, which indicate that blue light is harmful to cells in the body. But on the whole, the work is actually routine without deep investigation, and the experimental design lacks completeness, meanwhile the significance and novelty of this work are not enough.

Specific comments:

1. Cancer cells endure higher oxidative stress than normal cells due to their malignant transformation and metabolic disturbance. Mouse melanoma B16-F10 was selected for in vitro experiments to demonstrates the damage of blue light for cells in the body. The authors should indicate whether normal cells after blue light irradiation also exhibit increased ROS levels consistent with cancer cells.

2. UV and blue light have limited penetration depth in vivo because of substantial absorption by skin and underlying soft tissues. There are no enough animal experiments to verify that similar physiological effect also exhibit in vivo in this work.

3. Growing evidence indicated that blue light induce ROS generation for retina cells injury or DNA damage, the significance and novelty of this work are not enough.

4. Phenol red contained cell medium would affect the modulated effects of blue light to cells. In this paper, there are no elaborate the experimental details when blue light irradiation was performed.

5. In Figure 1D, Figure 3A and 3C, the fluorescence images of cells apoptosis, ROS generation and DNA damage are not clear enough. And the authors should provide the bright field images to demonstrated that no significant difference in cells number.

6. In Figure 2, there is no enough evidence supported that B16-F10 cells migrated after blue light irradiation. The author should explain more in detail.

7. In Figure 4, mRNA changes of several factors (TYR, TRP-1, TRP-2, mitf) were detected to show the reasons for melanin synthesis in B16-F10 cells. However, the protein level-changes are not being demonstrated, more tests and results should be shown here.

Reviewer #3: The reported work addresses the important and increasingly studied issue of the effect of visible light, and in particular blue light, on skin. Important contributions have been published in the field in the recent years. The novelty in the work by Zhou et al, performed on a melanocyte cell line, is the use of a semi-chronic exposure pattern with repeated irradiation at low fluences. It seems also that the authors used a 470 nm source, namely photons of lower energy than in other works. The text is yet not clear on this aspect. The emission spectrum of the lamp should be provided or at least described. The expression “low energy” should be rephrased because it can be understood as low energy photons or low fluence.

Using this irradiation protocol, the authors studied the phototoxicity of blue light, the induction of reactive oxygen species, the genotoxicity and the melanin synthesis. Major improvements have to be made on all these aspects.

The phototoxicity is studied by a MTT viability test, a trypan blue survival assay and cell counting. Exposures last 1 h or 2 h per day for 1, 2 or 3 days. Viability and cell counting lead to the same conclusion that the exposure protocol exhibit to a strong phototoxicity. In contrast, the trypan blue assay shows no impact on cell survival. These data are difficult to reconcile. The observation of a strong phototoxicity is a key point for the interpretation of all the other studied endpoints. It is worth mentioning that no apoptosis is detected; but no positive control is used that could establish the reliability of the technique used.

The next investigated response is a wound healing assay reflecting proliferation. The authors failed to observe migration of cells in the scratched area. The strong phototoxicity could well explain this result since dying cell will not replicate. The authors actually report a decrease in cell area, which shows more a cell death than a lack of migration.

The authors then report induction of ROS and formation of DNA damage revealed as double-strand breaks through the gamma-H2AX assay. Again, the fact that cells are dying in large proportion should make the authors more careful in their conclusion. They should also correct the discussion of DNA damage inducing oxidative stress. The succession of events is actually opposite.

The last part of the manuscript devoted to induction of melanogenesis by blue light is more convincing as it involves both quantification of melanin and assessment of the expression of a series of relevant genes.

In summary, this manuscript report some interesting results but the discussions and the conclusions have to be much more carefully made because of the phototoxicity under the irradiation condition. The authors also may want to be less alarmist on the health impact of their findings. Repeated reference to ocular toxicity is out of the scope of this study on skin. There is a big difference between cells in a plate and the human body. In that respect, the abstract should be rewritten with more scientific input and presentation of the present study.

6. PLOS authors have the option to publish the peer review history of their article (what does this mean?). If published, this will include your full peer review and any attached files.

Reviewer #1: No

Reviewer #2: No

Reviewer #3: No

---

## [Author Response · Author response to Decision Letter 0]

19 Dec 2022

Dear editor:

Thank you for giving us the opportunity to revise our manuscript for publication in PLOS ONE. We appreciate the time and effort that you and the reviewers dedicated to providing feedback on our manuscript and are grateful for the insightful comments on and valuable improvements to our paper.

We have incorporated most of the suggestions made by the reviewers. Those changes are highlighted in the manuscript. Please see below, in red, for a point-by-point response to the reviewers’ comments and concerns. All page numbers refer to the revised manuscript file with tracked changes.

Comments to the Author

Reviewer #1: 

Manuscript is well presented. This study provides important information that helps better understand the effect blue light has on cells. The abstract is clear and describes the background, goals, and results of the paper well. As a general comment, the grammar could be improved throughout the manuscript before the final submission. My specific comments are below:

INTRODUCTION

1) It may be beneficial to more clearly state the goals of this study. The last sentence of the introduction (page 4 line 51) states that “this study investigated the physiological effects of LED blue light on melanin synthesis in cells.” However, the paper examines many more things than just melanin synthesis. It also may be useful to provide more background in the introduction that will be later discussed in the paper, such as on wound healing, ROS, and DNA damage.

Replay:

Thank you very much for your comment. It is really that we didn’t state an accurate objective in our introduction. We inserted some more background contents about wound healing, ROS and DNA damage in the introduction as “Oxidative stress is associated with many cellular responses, one of which is DNA damage in cells [12]. UV exposure can cause single-strand or even double-strand DNA breaks [1], and UV exposure can also lead to an increase in intracellular oxidative stress [13]. In view of the fact that high-energy blue light can cause cell damage in previous studies, we believe that it is necessary to explore whether blue light with longer wavelength and lower energy density can lead to the increase of reactive oxygen species in cells and DNA damage. Many studies have shown that red light can promote the movement and proliferation of cells [5,6,14]. However, compared to the negative effects of blue light on cells in other studies, a recent study showed that 470nm blue light can also promote wound healing effect in rat model [14]. Therefore, we wanted to verify the effect of blue light with longer wavelength and lower energy density on the ability of cell migration.” Please refer to page 4 line 55-65.

And we also rewrote the objective as “In this study, we investigated how longer wavelength and low energy density blue light multipes exposure affect the physiological effect (phototoxicity, proliferation, migration, ROS generation, DNA damage and melanin synthesis) on B16F10 melanoma cells.”, please refer to page 5 line76-78. 

2) Page 3 Line 29 – The first sentence of the paper is confusing/ rather long, I would recommend to divide it into two sentences.

Reply:

Thank you very much for your advice, we rewrote this sentence from “The development of technology has increased the variety of light received in our daily lives, especially through the increasing popularity of television screens, computers, and smart phones, and more light is used late at night.” to “With the development of technology, we have to be exposed to more light in our daily life. Especially the longer television, computers, and smart phones use further extend the time and increase the frequency our exposure to visible light.”, please refer to page 3 line 35-37.

3) Page 3 line 39 – How is blue light significantly harmful to cells? Please provide more detail. 

Reply:

Thank you very much for your advice, the harmful effect or the negative effect of blue light to cells we have provided in the last half of this paragraph (page 3). We think the sentence “However, blue light with a wavelength of 400 nm to 495 nm is significantly harmful to cells, including retina cells” is not very appropriate. We revised this sentence to “However, research about blue light with a wavelength of 400 nm to 495 nm showed negative effects to cells. For example, 415-455 nm high energy blue light affects retina cells causing dry eye cataract and age-related macular degeneration [7].”, please refer to page 3 line 44-46. 

METHODS

1) MTT is an enzyme assay and not a viability assay, despite it often being mistakenly used as a viability assay. This may account for the discrepancy seen when comparing with trypan blue. Either the authors need to label the MTT as an enzyme function assay or remove from the manuscript.

Reply:

Thank you very much for your advice, we are agreeing with your comment that MTT assay is an enzyme assay. In our opinion, when there is no big difference on cell number, the MTT result can be also recognized as viability result. But this time, we found the decreased MTT result and the decreased cell number, so it was difficult to explain the decreased MTT was from the decreased cell number or the decreased enzymatic activity by blue light. Therefore, we further made the trypan blue assay and the proliferation assay. The no changed survival rate and inhibited cell proliferation were investigated. We also measured the cell cycle related protein mRNA level, but it was not a very proper method for cell cycle, and we didn’t have a flowcytometry, so we put this results in supplemental data for support our result that longer wavelength low energy blue light exposure inhibits the cell proliferation without affect the cell survival rate. But your advice is very helpful for us to make a better essay. We revised the “cell viability” in this manuscript to “cell enzymatic activity”. Please refer to method page 6 line 102 and results page 9 line171. 

2) Page 5 line 75 – Please be consistent with the use of “day” and “d” throughout.

Reply:

Thank you very much for your advice, we revised all the “d” to “days” in all the manuscript.

3) Page 6 line 96 – How many hours each day were the cells treated with blue light?

Reply:

Thank you very much for your question, we found we forgot to mention the exposure time in each day. We inserted the treatment time for each day as “cells that were treated with blue light (0, 1 or 2 h each day) for 3 days”, please refer to page 7 line 120. 

4) Page 6 line 101 – Please change “confluence” to “confluent.”

Reply:

Thank you very much for your advice, we revised the confluence to confluent, please refer to page 7 line 125.

RESULTS

1) Perhaps repeat the experiments in an additional cell line to ensure that these findings are not due to single cell line variability and are conserved across cell lines.

Reply:

Thank you very much for your suggestion, we think it is a very good suggestion indeed. While carrying out experiments on B16F10 melanoma cells, we also performed similar experiments on other cell lines. Due to the special feature of melanin synthesis by B16F10, we made a separate article. Similarly, we also exposed other cells (such as muscle cells, fibroblasts, and mesenchymal stem cells) under the same conditions. In some experiments, we also got the same results as melanocytes. But we hope to test the properties of other special cell lines after blue light irradiation, so we will discuss this in more detail in a future article.

2) If blue light did not induce apoptosis, why was proliferation limited? The authors should add cell cycle experiments to determine blue light's effect on cell cycle. Additionally, is it possible that the cells were necrotic or fragmented and not picked up on flow cytometry?

Reply:

Thank you very much. As our reply in the “Method (1) about MTT”, we also measured the cell cycle related factors mRNA level. We measure the mRNA level of cells after irradiation and put in dark 3 hours, cyclin A, B and E was down regulated by the blue light irradiation. This data we will add to supplemental data to support our conclusion and add the description in the discussion, as “We also detected the cyclin level after light exposure (Fig. S1), the decreased cyclin mRNA also supports that the cell cycle was arrested by blue light.” please refer to page 14 line 274-276. 

Further, the kit for apoptosis we used also can detect cell necrosis, and there were no necrosis cells be detected in our results (right side). Therefore, we didn’t put the data in our manuscript. 

3) Throughout the discussion and in the figure descriptions, authors did not say if findings were significant for numerous results (example – page 9 line 169, wound healing assay). Please always state if results are statistically significant.

Reply:

Thank you very much for your advice. We checked the article and made some revisions to the results section to clearly point the findings were significant or not. “We found that the cells appeared in the space area after 18 hours was decreased” at page 9 line 169 was changed to “We found that the area changes of the cross area without cells within 18 hours significantly decreased due to the blue light 1- and 2- hours irradiation (Figure 2)” Page 10 line 196. We also inserted the statement at Page 11 line 212-213 as “It is clear that the green fluorescence was increased in the blue light exposure groups (Figure 3A), the results by microplate reader also accordance with the microscope that blue light significantly increased the fluorescent in cells (Figure 3B)” in ROS generation results; and at Page 12 line 239-240 as “low energy density multiple irradiation also slightly (about 1.2-1.3 times) but significantly promoted melanin synthesis” in melanin contents results.

4) How was the DNA damage assay selected? 

Reply:

Thank you very much for your question. Because UV can cause DNA damage, the rise of ROS can also cause DNA damage, and we detected the rise of ROS by blue light irradiation. Therefore, we also wanted to investigate whether blue light would cause a same result.

5) Page 8 line 151 – Authors state “longer blue light irradiation times reduced cell viability.” Please clarify how exactly long cells were irradiated in this result. And was this reduction in cell viability statistically significant?

Reply:

Thank you very much for your comment. The sentence about “longer blue light irradiation time” was really confusing, and also to clearly descript the statistically significant we revised this part from “MTT experiments showed that longer blue light irradiation time reduced cell viability (Figure 1A).” to “MTT assay showed that longer wavelength and low energy density blue light irradiation significantly reduced cell enzymatic activity (Figure 1A), even only once irradiation the result of MTT was also decreased.”, please refer to page 9 line 171-173. 

6) Page 9 line 159 (Fig 1C) – Clarify which assay was used in the figure description.

Reply:

Thank you very much for your comment. Here we directly counted the cell number by a cell counter, so we inserted the cell counter to the figure description. “C. Cell were irradiated once, returned to the dark incubator, and cell numbers were counted by cell counter every following day.” Please refer to page 10 line 186.

7) Page 10 line 181– The first sentence in the ROS paragraph might fit better in the introduction.

Reply:

Thank you very much for your comment. As a background for ROS and DNA damage, we added this description into the introduction as “Oxidative stress is associated with many cellular responses, one of which is DNA damage in cells [12]. UV exposure can cause single-strand or even double-strand DNA breaks [1], and UV exposure can also lead to an increase in intracellular oxidative stress [13].”, please refer to page 4-5 line 54-56.

8) Page 10 line 185 – Was intracellular ROS significantly increased?

Reply:

Thank you very much for your comment. The result of ROS is significant, we inserted the description as “It is clear that the green fluorescence was increased in the blue light exposure groups (Figure 3A), the results by microplate reader also accordance with the microscope that blue light significantly increased the fluorescent in cells (Figure 3B).”, please refer to page 11 line 212-213. 

9) Page 10 line 190 – State what kit was used to measure DNA damage.

Reply:

Thank you very much for your comment. “DNA Damage Detection Kit - γH2AX　– Green” was used to measure DNA damage. We added the name of the kit at page 11 line 219 as “measured the DNA damage with the “DNA Damage Detection Kit - γH2AX - GREEN” kit”

10) Page 11 line 206 – Again, the first couple sentences of this paragraph belongs in the introduction, as this is the background/ reasoning for conducting these experiments

Reply:

Thank you very much for your comment. This statement we also added into the introduction as “Previously, short wave high energy blue light has been found able to promote the melanin synthesis in melanocytes through up-regulating tyrosinase and MITF [20, 21]”, please refer to page 4 line 71-74. 

11) Page 11 line 209 – Authors state, “irradiation also slightly promoted melanin synthesis.” What does “slightly” mean?

Reply:

Thank you very much for your comment. We revised this part to “low energy density multiple irradiations also slightly (about 1.2-1.3 times) but significantly promoted melanin synthesis”, please refer to page 12 line 239. 

DISCUSSION/ CONCLUSION

I believe this section could discuss the results of the experiments more. What are some possible reasons why low energy blue light inhibits migration? What is the significance of low energy blue light causing increased ROS and DNA damage to our cells? What are some potential reasons the author’s study found that blue light inhibits cell migration, when it is reported in other studies that ROS promotes cell migration?

Reply:

Thank you very much for your comment. We made a further discussion about our results.

Firstly, for the blue light inhibit migration and the result that increased ROS and inhibited cell migration. We learned from the paper that ROS, especially H2O2, plays an important role on cell migration. But an increased ROS will also induce DNA damage, further cause the cell cycle arrest and DNA repair. Therefore, the migration inhibition and increased ROS are not contradictory. We inserted this in the discussion section “ROS also play an important role in cells，appropriate increased of hydrogen peroxide (H2O2) will promote intracellular signal conduction and thus promote cell migration [30, 31]. However, excessive ROS production will produce corresponding toxicity [32]. Thus, we think the type of ROS up-regulated by blue light was not H2O2, and the increase level was also higher than the necessary. For this reason, the rise of ROS and the DNA damage by blue light irradiation, caused the cell cycle arrest, furtherly negatively regulated the cell migration ability.”, please refer to page 15 line 303-309. 

Secondly, about DNA damage and ROS generation, we also made a further discussion, “In our results, both ROS generation and DNA damage were up-regulated by blue light exposure, which suggested that although low energy density blue light didn’t induce apoptosis, it also induce the oxidative stress and DNA damage. ROS regulates many intracellular activities, while excess ROS induces intracellular damage, including DNA damage [12]. DNA damage caused by ROS is divided into single-strand break (SSB) and double-strand break (DSB) [28]. DNA damage also causes cell cycle arrest and DNA repair [12]. Severe double-strand break of DNA damage directly leads to cell apoptosis [29]. These results help us understand why blue light causes DNA damage and reduces cell proliferation without causing apoptosis. We think the reason is the relatively low energy density blue light used in our experiments, the energy isn’t stronge enough to cause fatal damage to the cells. These results also remind us that we need to pay more attention to the daily use of digital device. Although there will be no serious harm, continuous long-term exposure will also lead to an increase in oxidative stress and a decrease in the proliferation of skin cells”, please refer to page 14-15 line 285-297.

How do the results of this study compare to other studies? The work here is very nice, and similar to the works referenced in manuscript reference #20 by Mamalis et al. Please note how you distinguished your works.

Reply:

Thank you very much for your comment. As your comment, our works were indeed very similar to Mamalis et al. in terms of experimental content. But we focused on the melanin synthesis cells in the epidermis in this experiment. And different from most of the experiments, we used blue light with a longer wavelength (470nm) and used relatively lower irradiation energy for several times of irradiation, which was used to mimic the daily use. We didn’t explicitly emphasize this point at the beginning, so we added these points in the introduction as “However, in most of these studies, shorter wavelength (400-450 nm) and higher energy blue light is used, which is unusual to simulate the situation in our daily life. Therefore, we wondered how longer wavelength (470 nm) lower energy density and multiple exposure of blue light affect the physiological function of cells.” at page 3 line 50-53.

The authors also need to add more to the limitations section and how they mitigated against these challenges.

Reply:

Thank you very much for your comment. 

We added the limitation about the ROS and DNA damage at “Since we detected ROS and DNA damage at the same time point after light exposure in this experiment, it is still difficult to determine whether the DNA damage is directly caused by irradiation or by the ROS generated after light exposure. This point needs to be verified by further experiments in the future.”, please refer to page 15 line 298-301

We also added the limitation about the ROS and migration at “However, it is still necessary to identify the type of ROS production induced by blue light irradiation (H2O2 promotes cell migration). Therefore, further studies are required to investigate how blue light inhibits cell migration and determine the relationship between blue light induced ROS generation and inhibited migration.”, please refer to page 15-16 line 309-313. 

We also added the limitation about the in vivo experiment at page 16 line 333-338, “Last but not least, different wavelength of visible light has different penetration to the skin, and there is also a big difference between cells in a plate and the body. Also, the wound healing assay on rat previously suggested that both red and blue light irradiation promoted the wound healing [14], which was different with our results on migration. In order to investigate the function of blue light to the live skin and verify the depth with impact of blue light, more rigorous in vivo. experiments are still required.”

Additionally, the conclusion of the paper mostly restates the results. Are there any other final thoughts or conclusions that the authors would like readers to take away from this paper?

Reply:

Thank you very much for your comment. We revised the conclusion section to “Multiple exposure to longer wavelength (470nm) low energy density (1 or 2 J/m2) blue light didn’t induce cell apoptosis but still inhibited the cell proliferation. This effect was induced by the increased ROS generation and DNA damage. But it is still necessary to investigate the mechanism how blue light induced the DNA damage and ROS generation. Longer wavelength low energy density multiple blue light exposure can also promote melanin synthesis. These results remind us that even low energy density blue light has a weak phototoxicity, repeated exposure will also cause cellular damage and regulate cellular function. It is better to pay more attention to the daily using time and frequency of the digital device to protect us with the damage from blue light”, please refer to page 17 line 341-349.

FIGURES

Figure 1C – This graph does not show if the results are statistically significant.

Reply:

Thank you very much for your comment. We revised the graph to show the statistically significant more clearly. 

Reviewer #2: 

In this paper, low energy and multiple blue light irradiations were used to promote melanin synthesis and DNA damage in mouse melanoma B16-F10 cells, which indicate that blue light is harmful to cells in the body. But on the whole, the work is actually routine without deep investigation, and the experimental design lacks completeness, meanwhile the significance and novelty of this work are not enough.

Specific comments:

1. Cancer cells endure higher oxidative stress than normal cells due to their malignant transformation and metabolic disturbance. Mouse melanoma B16-F10 was selected for in vitro experiments to demonstrates the damage of blue light for cells in the body. The authors should indicate whether normal cells after blue light irradiation also exhibit increased ROS levels consistent with cancer cells.

Reply:

Thank you very much for your comment. We agree with your opinion about the ROS increased. We have also performed irradiation of blue light on other cell lines using the same conditions. For example, we used mesenchymal stem cells, which also showed an increase in ROS after blue light exposure (figure at right side). However, we have other plans for the research results on light and mesenchymal stem cells, which will be used in future papers, so we cannot present them in this manuscript. 

2. UV and blue light have limited penetration depth in vivo because of substantial absorption by skin and underlying soft tissues. There are no enough animal experiments to verify that similar physiological effect also exhibit in vivo in this work.

Reply:

Thank you very much for your comment. We also agree with your point of view, it is necessary to perform an in vivo. experiment to confirm the effect of blue light. But due to covid-19 and the relocation of our laboratory, it is difficult for us to complete animal experiments in a short time. We will add this as a limitation of our research in discussion section as “Last but not least, different wavelength of visible light has different penetration to the skin, and there is also a big difference between cells in a plate and the body. Also, the wound healing assay on rat previously suggested that both red and blue light irradiation promoted the wound healing [14], which was different with our results on migration. In order to investigate the function of blue light to the live skin and verify the depth with impact of blue light, more rigorous in vivo. experiments are still required.” at page 17 line 333-338, and hope this will be more enlightening for readers of the paper. And we will restart follow-up experiments after we settle down.

3. Growing evidence indicated that blue light induce ROS generation for retina cells injury or DNA damage, the significance and novelty of this work are not enough.

Reply:

Thank you very much for your comment. By introducing the generally accepted view that blue light causes retinal cell damage, we initially wanted to evoke the thinking how blue light affects somatic cells. As cells with photoreceptor, retinal cells may not respond exactly the same to somatic cells when exposed to light. From our results, it was found that even in the absence of photoreceptors, blue light with longer wavelength and weaker energy can cause cell damage and increase ROS. This may indicate that light can affect cell function and survival even without passing through the photoreceptors. We respect your point of view, but to expand on this topic, we don't have enough experimental data to explain the difference between somatic and retinal cells. And the ocular toxicity is out of the scope of this study on skin. Therefore, we decided to remove the emphasizing and discussion of retinal cells from the article at old version page12 line 229-231 “Exposure to blue visible light is harmful to most cells, especially retinal cells [7]. Therefore, the blue light-blocking function was added in eyeglasses or screen displays to reduce the damage caused by blue light to retinal cells [18,19].”, and only use it as a background for the regulation of cell function by blue light. “However, blue light with a wavelength of 400 nm to 495 nm is significantly harmful to cells, including retina cells [7]” in old version introduction page 3 line 36, was changed to “However, research about blue light with a wavelength of 400 nm to 495 nm showed negative effects to cells. For example, 415-455 nm high energy blue light affects retina cells causing dry eye cataract and age-related macular degeneration [7].”, please refer to page 3 line 44-46. 

4. Phenol red contained cell medium would affect the modulated effects of blue light to cells. In this paper, there are no elaborate the experimental details when blue light irradiation was performed.

Reply:

Thank you very much for your comment. To avoid the effect from phenol red, HBSS without phenol red was used to instead of medium when irradiating. We thought the description in the method maybe not detailed enough, so we revised it as “HBSS (Wako, Japan) without phenol red was used in place of the medium during blue light treatment to prevent stimulating effects on cells. After blue light irradiation, HBSS was replaced with fresh RPMI-1640 medium and put back to dark incubator. The non light group was changed to HBSS or refreshed to medium together with irradiation groups, but the dish was kept in dark incubator.”, please refer to page 5-6 line 95-99. 

5. In Figure 1D, Figure 3A and 3C, the fluorescence images of cells apoptosis, ROS generation and DNA damage are not clear enough. And the authors should provide the bright field images to demonstrated that no significant difference in cells number.

Reply:

Thank you very much for your comment. Regarding the not clear photos, we directly used the shrink original image to avoid the loss of details. Because the image become too small, the image didn’t very clear. We have enlarged and cut the picture so that the picture does not look so small (show on right side). 

For bright field cells, in Figure 1 we used cells after three days of irradiation. From Figure 1C, we can find that after three days, there was a significant difference in the cell number. For the apoptosis kit, it is also available to detect the alive cells, so we inserted the alive cell field and overlay.

In Figure 3, we stained directly after illumination, and there was no difference in cell density. We showed the brightfield images in the supplemental figure S3 and S4. Please refer to page 12 line 231

6. In Figure 2, there is no enough evidence supported that B16-F10 cells migrated after blue light irradiation. The author should explain more in detail.

Reply:

Thank you very much for your comment. At the beginning, we draw the black cross line to show the space without cells after makes the gap. The photos after 18 hours, the same cross with 0 hour was pasted to show the movement of the cells (the red area was decreased as the blue light irradiation). 

But for the migration assay, the lines were usually used to directly label the area without cells, so that the change of the area of the gap means the migration of cells. To make our results more easily to understand, we change the form for cross line as the normal way shown at right side, please refer to figure 2A. We also added some description to help understand the figures as “The yellow line was used to label the cross area without cells. We found that the area changes of the cross area without cells within 18 hours significantly decreased due to the blue light 1- and 2-hours irradiation (Figure 2),”, please refer to page 10 line 195-196.

7. In Figure 4, mRNA changes of several factors (TYR, TRP-1, TRP-2, mitf) were detected to show the reasons for melanin synthesis in B16-F10 cells. However, the protein level-changes are not being demonstrated, more tests and results should be shown here.

Reply:

Thank you very much for your comment. We also believe that the protein level-changes are very important for explaining how blue light regulates melanin synthesis. Before detecting the mRNA content, we have made dozens of attempts on the protein level at different time points. But unfortunately, we couldn't find a stable change tendency of a protein level. In desperation, the mRNA level by real-time q-PCR was found that stable changes. Since much is unknown about how blue light affects cellular metabolism, we will add this to the limitations of the experiments in the discussion as “We also detected the protein levels of tyrosinase, MITF, TRP-1 and TRP-2. Unfortunately, although we tested the samples after 0, 3, 12, 24, and three times of irradiation, it was still unable to get a stable change of the protein content. We suspect that this is due to disruption of cellular protein synthesis function due to DNA damage. Thus, it is necessary to conduct more research to investigate the mechanism and the target of blue light regulating melanin synthesis.”, please refer to page 16 line 320-325.

Reviewer #3: 

The reported work addresses the important and increasingly studied issue of the effect of visible light, and in particular blue light, on skin. Important contributions have been published in the field in the recent years. The novelty in the work by Zhou et al, performed on a melanocyte cell line, is the use of a semi-chronic exposure pattern with repeated irradiation at low fluences. It seems also that the authors used a 470 nm source, namely photons of lower energy than in other works. The text is yet not clear on this aspect. The emission spectrum of the lamp should be provided or at least described. The expression “low energy” should be rephrased because it can be understood as low energy photons or low fluence.

Replay:

Thank you very much for your comment, we are so sorry that we didn’t mention the wavelength of the blue light, we added the wavelength of the blue light in the method section as “The blue LED light (peak at 470nm, DC24V, 1.13A, Max 27.2W) used in this experiment is a product of AITECSYSTEM CO., LTD. (Japan)”, please refer to page 5 line 90. As your view, the “low energy” blue light was not accurate, we have rewritten it to “longer wavelength low energy density”.

Using this irradiation protocol, the authors studied the phototoxicity of blue light, the induction of reactive oxygen species, the genotoxicity and the melanin synthesis. Major improvements have to be made on all these aspects.

The phototoxicity is studied by a MTT viability test, a trypan blue survival assay and cell counting. Exposures last 1 h or 2 h per day for 1, 2 or 3 days. Viability and cell counting lead to the same conclusion that the exposure protocol exhibit to a strong phototoxicity. In contrast, the trypan blue assay shows no impact on cell survival. These data are difficult to reconcile. The observation of a strong phototoxicity is a key point for the interpretation of all the other studied endpoints. It is worth mentioning that no apoptosis is detected; but no positive control is used that could establish the reliability of the technique used.

Replay:

Thank you very much for your comment. Typically, significant cytotoxicity induces many dead cells floating in the medium. However, in this experiment, no floating cells were found in the medium, which is why we would like to use trypan blue to further measure the survival rate of the cells. The results of trypan blue show that although 470nm low energy density blue light irradiation has a phototoxicity, it is not the kind of severe toxicity that directly kills cells, but a slow and slight toxicity. The blue light exposure induced damage, while the cells still alive. We are also very grateful to you for the inspiration of this sentence “The observation of a strong phototoxicity is a key point for the interpretation of all the other studied endpoints.”, which gave us a very good idea to combine our results. About this, we will explain in more detail below.

For the apoptosis assay, we also provided the alive cells field in revised Fig 1D, and also provided the protein level of cleaved-caspase-3 in supplemental figure 2 (right side) to support the result of no apoptosis cells. 

The next investigated response is a wound healing assay reflecting proliferation. The authors failed to observe migration of cells in the scratched area. The strong phototoxicity could well explain this result since dying cell will not replicate. The authors actually report a decrease in cell area, which shows more a cell death than a lack of migration.

The authors then report induction of ROS and formation of DNA damage revealed as double-strand breaks through the gamma-H2AX assay. Again, the fact that cells are dying in large proportion should make the authors more careful in their conclusion. They should also correct the discussion of DNA damage inducing oxidative stress. The succession of events is actually opposite.

Replay:

Thank you very much for your comment, these two parts we want to explain together. We agree with you that our conclusions suggest that blue light is phototoxic, and this toxicity leads to a reduction in proliferation. However, we did not detect cell death (floating cells in medium). We thought the blue light we used has a very low energy, which will not produce severe cytotoxicity and even directly kill cells like UV. Because of the lower energy, the resulting DNA damage and ROS rise are still within the survivable range of cells. It has been investigated that the rise of ROS will cause DNA damage, the DNA damage will further lead to the cell cycle arrest and DNA repair. This is also consistent with our results that "After irradiation, ROS increase and DNA damage induced the inhibition of cell proliferation". It is thanks to your previous sentence that inspired us. Therefore, we can conclude that blue light with longer wavelength and lower energy density, although not directly killing cells, is still phototoxic and will affect the physiological functions of cells. About this we added it to discussion section as “In our results, both ROS generation and DNA damage were up-regulated by blue light exposure, which suggested that although low energy density blue light didn’t induce apoptosis, it also induce the oxidative stress and DNA damage. ROS regulates many intracellular activities, while excess ROS induces intracellular damage, including DNA damage [12]. DNA damage caused by ROS is divided into single-strand break (SSB) and double-strand break (DSB) [28]. DNA damage also causes cell cycle arrest and DNA repair [12]. Severe double-strand break of DNA damage directly leads to cell apoptosis [29]. These results help us understand why blue light causes DNA damage and reduces cell proliferation without causing apoptosis. We think the reason is the relatively low energy density blue light used in our experiments, the energy isn’t stronge enough to cause fatal damage to the cells. These results also remind us that we need to pay more attention to the daily use of digital device. Although there will be no serious harm, continuous long-term exposure will also lead to an increase in oxidative stress and a decrease in the proliferation of skin cells.”, please refer to page 14-15 line 286-298.

For，the migration assay. At the beginning, we draw the black cross line to show the space without cells after makes the gap. The photos after 18 hours, the same cross with 0 hour was pasted to show the movement (or the growth) of the cells (the red part in the photo right side). 

However, the lines in photos are usually used to directly label the area without cells, so that the change of the area of the gap means the migration of cells. To make our results more easily to understand, we change the form for cross line as the normal way like shown at right side, please refer to figure 2A. We also added some description to help understand the figures as “The yellow line was used to label the cross area without cells. We found that the area changes of the cross area without cells within 18 hours significantly decreased due to the blue light 1- and 2-hours irradiation (Figure 2),”, please refer to page 10 line 195-196.

About the succession of events of ROS increase and DNA damage, our conclusion was a little bit careless. Therefore we revised and added in discussion section as “Since we detected ROS and DNA damage at the same time point after light exposure in this experiment, it is still difficult to determine whether the DNA damage is directly caused by irradiation or by the ROS generated after light exposure. This point needs to be verified by further experiments in the future.”, please refer to page 15 line 299-302.

The last part of the manuscript devoted to induction of melanogenesis by blue light is more convincing as it involves both quantification of melanin and assessment of the expression of a series of relevant genes.

In summary, this manuscript report some interesting results but the discussions and the conclusions have to be much more carefully made because of the phototoxicity under the irradiation condition. The authors also may want to be less alarmist on the health impact of their findings. Repeated reference to ocular toxicity is out of the scope of this study on skin. There is a big difference between cells in a plate and the human body. In that respect, the abstract should be rewritten with more scientific input and presentation of the present study.

Replay:

Thank you very much for your comment to our abstract, we recognized the inadequacy of our abstract. We have rewritten the abstract to 

“Visible light is present everywhere in our lives. Widespread use of computers and smartphones has increased the daily time spent in front of screens. What effect does this visible light have on us? Recent studies have shown that short-wavelength blue light (400-450nm) irradiation, similar to UV, inhibits the cell proliferation and differentiation, induces the intracellular oxidative stress, promotes the cell apoptosis and causes some other negative effects. However, it’s unusual that directly face to such short-wavelength and high-energy blue light in daily life. Therefore, the effects of blue light with longer wavelength (470nm), lower energy (1, 2 J/cm2) and multiple times (simulated daily use) exposure on cells have been studied in this experiment. In our results, low energy density multiple blue light inhibited cell proliferation and metastatic capability with a weak phototoxicity. Blue light also promoted intracellular reactive oxygen species, and caused DNA damage. Furthermore, the melanin synthesis was also promoted by low energy density multiple blue light exposure. Together, these results indicate that longer wavelength and low energy density blue light multiple exposure is still harmful to our cells even the effect is lower than short wavelength high energy blue light. Furthermore, prolonged exposure to screens likely induces dull skin through induction of melanin synthesis. These results further mentioned us should paid more attention to controlling the daily use of digital device.”, please refer to page 2 line 14-30.

---

## [Decision Letter · Decision Letter 1]

4 Jan 2023

PONE-D-22-27769R1Low Energy Multiple Blue light-emitting diode light Irradiation promotes melanin synthesis and induces DNA damage in B16F10 melanoma cellsPLOS ONE

Dear Dr. Sakamoto,

Thank you for submitting your manuscript to PLOS ONE. After careful consideration, we feel that it has merit but does not fully meet PLOS ONE’s publication criteria as it currently stands. Therefore, we invite you to submit a revised version of the manuscript that addresses the points raised during the review process.

ACADEMIC EDITOR: Please consider the comments from reviewer 1 ==============================

We look forward to receiving your revised manuscript.

Kind regards,

Yi Cao

Academic Editor

PLOS ONE

Journal Requirements:

Reviewers' comments:

Reviewer's Responses to Questions

**Comments to the Author**

1. If the authors have adequately addressed your comments raised in a previous round of review and you feel that this manuscript is now acceptable for publication, you may indicate that here to bypass the “Comments to the Author” section, enter your conflict of interest statement in the “Confidential to Editor” section, and submit your "Accept" recommendation.

Reviewer #1: (No Response)

Reviewer #2: All comments have been addressed

2. Is the manuscript technically sound, and do the data support the conclusions?

Reviewer #1: Yes

Reviewer #2: Yes

3. Has the statistical analysis been performed appropriately and rigorously? 

Reviewer #1: Yes

Reviewer #2: Yes

4. Have the authors made all data underlying the findings in their manuscript fully available?

Reviewer #1: Yes

Reviewer #2: Yes

5. Is the manuscript presented in an intelligible fashion and written in standard English?

Reviewer #1: No

Reviewer #2: Yes

6. Review Comments to the Author

Reviewer #1: Title: Low Energy Multiple Blue light-emitting diode light Irradiation promotes melanin synthesis and induces DNA damage in B16F10 melanoma cells

Authors: Siqi ZHOU, Ryusuke YAMADA, Kazuichi Sakamoto

This paper provides important information that helps better understand the effect blue light has on cells. The abstract is clear and describes the background, goals, and results of the paper well.

As a general comment, the English grammar needs to be improved throughout the manuscript before the final submission. My specific comments are below:

Introduction

1. It may be helpful to provide further background and statistics on the prevalence of screen use in recent years, especially due to COVID. It would also be helpful to clearly specify what wavelength and energy of blue light is present in most daily use screens.

2. Authors discuss red light in the second paragraph of the introduction. It would be helpful to provide further background on the effect red light has on cellular proliferation, ROS, and DNA damage, as these are things this paper is studying.

Methods

1. Please specific the light irradiation protocol (ex/ how many hours of irradiation per day) in the “Light irradiation” section of the introduction.

2. What plating density was used for the light irradiation?

3. Please specify how long after light irradiation each assay was performed.

Discussion

1. What is the significance of the fact that blue light inhibited cellular proliferation, but did not cause apoptosis? (page 14)

2. Perhaps more clearly discuss why the MTT assay showed reduced cell enzymatic activity, but the trypan blue assay shows no impact on cell survival in the discussion.

3. What is the significance of blue light inhibiting cell migration? (page 15)

4. What do authors mean when they state “intracellular ROS and DNA damage immediately occurred after blue light irradiation, and there was no significant change after three days irradiation? (page 16 line 327). In the methods and results, authors only discussed that immediately after blue light irradiation, ROS production and DMA damage was increased.

5. In the abstract, authors state “low energy density blue light multiple exposure is still harmful to our cells even though the effect is lower than short wavelength high energy blue light.” Please cite evidence of this and discuss in the discussion.

Figures

1. Please define all abbreviations used in figures in the key (ex/ NL, B1h, B2h)

Reviewer #2: (No Response)

7. PLOS authors have the option to publish the peer review history of their article (what does this mean?). If published, this will include your full peer review and any attached files.

Reviewer #1: No

Reviewer #2: No

---

## [Author Response · Author response to Decision Letter 1]

16 Jan 2023

Dear editor:

Thank you for giving us the opportunity to revise our manuscript for publication in PLOS ONE. We appreciate the time and effort that you and the reviewers dedicated to providing feedback on our manuscript and are grateful for the insightful comments on and valuable improvements to our paper.

We have incorporated most of the suggestions made by the reviewers. Those changes are highlighted in the manuscript. Please see below, in red, for a point-by-point response to the reviewers’ comments and concerns. All page numbers refer to the revised manuscript file with tracked changes.

Reviewer #1: 

Title: Low Energy Multiple Blue light-emitting diode light Irradiation promotes melanin synthesis and induces DNA damage in B16F10 melanoma cells

Authors: Siqi ZHOU, Ryusuke YAMADA, Kazuichi Sakamoto

This paper provides important information that helps better understand the effect blue light has on cells. The abstract is clear and describes the background, goals, and results of the paper well.

Reply: Thank you very much.

As a general comment, the English grammar needs to be improved throughout the manuscript before the final submission. My specific comments are below:

Introduction

1. It may be helpful to provide further background and statistics on the prevalence of screen use in recent years, especially due to COVID. It would also be helpful to clearly specify what wavelength and energy of blue light is present in most daily use screens.

Reply: Thank you very much for your comments. We agree with your point of view very much, so that we have added the relevant descriptions about COVID and the usage of digital device in our introduction section as “A recent web-based survey indicated that approximately 93.8% of respondents reported that their digital device usage hours increased under the influence of COVID. In particular, the student population saw an increase of 5.18 ± 2.89 hours per day of screen-facing time due to e-learning.”, please refer to page 3 line 37 – 40.

For the wavelength and energy of blue light is present in most daily use screens, we also think it is very important condition. So we added the wavelength of our daily used digital device screens as “Therefore, it is important to investigate the effect of light from the LED screen (there is a peak of light intensity in the 400-500 nm range of the screen spectrum) on our bodies and cells” at page 3 line 42-43. 

However, depending on the difference of devices, the duration (time) and the posture (distance) we used, the energy we got were different. Therefore, we are sorry that it is difficult to include a range of energy for daily use.

2. Authors discuss red light in the second paragraph of the introduction. It would be helpful to provide further background on the effect red light has on cellular proliferation, ROS, and DNA damage, as these are things this paper is studying.

Reply: Thank you very much. We have already provided the effect of red light on cell proliferation. We also searched the previous studies about the effect of red light on oxidative stress and DNA damage. These contents have been inserted into the introduction section as “And red light exposure does not cause oxidative stress and DNA damage”, please refer to page 3 line 47.

Methods

1. Please specific the light irradiation protocol (ex/ how many hours of irradiation per day) in the “Light irradiation” section of the introduction.

Reply: Thank you very much. We have revised the irradiation protocol to “For the 3-day experiment, the cells were exposed to blue light for 0, 1, and 2 hours per day and then returned back to the incubator (dark). After the irradiation of day 3, the cells were returned to the dark incubator for 24 hours before subsequent assays.”, please refer to page 6 line 99-101. 

2. What plating density was used for the light irradiation?

Reply: Thank you very much. The plating density of cells was 1.0 × 105 cells/ well in 60 mm dish, 3.0 × 103 cells/well in 24-well-plates, or 1.0 × 103 cells/well in 96-well-plates. We have mentioned this in the method section cell and cell culture (in page 5 line 90). We also checked and revised the description in each method of assay as: 

“Cells were cultured in 24-well-plates as a density of 3.0 × 103 cells/well” in the method of MTT assay, 

“Cells were cultured in 24-well-plates as a density of 3.0 × 103 cells/well.” in the method of Trypan blue assay and proliferation assay, 

“Cells were prepared in a Nunc black wall clear flat bottom 96 well plate (Thermo Fisher Scientific, USA) as a density of 1.0 × 103 cells/well.” in the method of apoptosis assay and ROS generation assay.

“1.0 × 105 cells were seeded into 6 cm dishes until confluent.” in the method of wound healing assay.

“Cells for melanin content were prepared in a 6-well-plate as a density of 5.0 x 104 cells/well” in the method of melanin content assay.

“Cells were prepared in 6 cm dishes as a density of 1.0 × 105 cells/dish” in the method of real time PCR

3. Please specify how long after light irradiation each assay was performed. 

Reply: Thank you very much. We have checked and revised the time after irradiation for the method of each assay. 

“The enzymatic activity was measured daily after the irradiation 24 hours.” in the method of MTT assay,

“The green fluorescence (Ex/Em = 490/525 nm) for apotosis cells and the blue (Ex/Em = 405/450 nm) for alive cells were stained after the day 3 irradiation 24 hours” in the method of apoptosis assay,

“Intracellular reactive oxygen species (ROS) were immediately measured after the irradiation with a DCFDA/H2DCFDA Cellular ROS assay kit” in the method of ROS generation,

“After day 3 irradiation, cells were returned to the dark incubator for 24 hours and then collected with trypsin,” in the method of melanin content assay,

“mRNA (immediately and 3 hours after blue light irradiation) was extracted using RNA iso Plus” in the method of Real time PCR.

Discussion

1. What is the significance of the fact that blue light inhibited cellular proliferation, but did not cause apoptosis? (page 14)

Reply: Thank you very much for your comment. Unfortunately, according to our results, it was not enough to explain the significance of these phenomenon very well. We thought this will be a very interesting topic in the future. We added some statement in our discussion part to show our inadequacy.” Longer wavelength lower energy blue light irradiation inhibited cell proliferation without inducing apoptosis. This is a very interesting phenomenon. To cancer treatment, it means that it is possible to inhibit cancer growth without killing the cells. While further experiments are still necessary to verify this conjecture.” Please refer to page 15 line 301-305.

2. Perhaps more clearly discuss why the MTT assay showed reduced cell enzymatic activity, but the trypan blue assay shows no impact on cell survival in the discussion.

Reply: Thank you very much for your comment. We discussed this point in discussion section more clearly as “Initially, we found the blue light irradiation decreased the cell enzymatic activity by MTT assay but didn’t affect the cell survival rate by trypan blue assay. Since the MTT results were based on both cell number and enzymatic activity, which a decreased cell number was also found. It was difficult to determine whether the decrease in MTT was due to a decrease in cell number or a decrease in cell enzymatic activity. Therefore, we further performed trypan blue and cell proliferation assays. Intermittent, multiple irradiations didn’t affect the cell survival rate but significantly reduced cell proliferation. These results indicated that the decreased enzymatic activity by blue light due to the reduced cell proliferation rather than cytotoxicity.”, please refer to page 15 line 287-294. 

3. What is the significance of blue light inhibiting cell migration? (page 15)

Reply: Thank you very much for your comment. As B16F10 melanoma cells were metastatic cells, the inhibition for migration maybe can give us some idea about blue light exposure inhibits the melanoma metastasis (benefit). The inhibition effect on cells migration also gave us ideas about blue light exposure slow down the wound healing (harm). Compared to other previous studies, blue light usually inhibits proliferation and metastasis. Therefore, we believe that our result of migration inhibition effect is still mainly indicative of the harmful effects of blue light. Therefore, we summarized this migration inhibition effect, together with DNA damage, proliferation inhibition as a weak phototoxicity in conclusion. “low energy density blue light has a weak phototoxicity, repeated exposure will also cause cellular damage and regulate cellular function” please refer to page 18 line 370.

4. What do authors mean when they state “intracellular ROS and DNA damage immediately occurred after blue light irradiation, and there was no significant change after three days irradiation? (page 16 line 327). In the methods and results, authors only discussed that immediately after blue light irradiation, ROS production and DMA damage was increased.

Reply: Thank you very much for your comment. At the beginning of ROS and DNA damage assay, we also detected the cells after 3 days treatment (the same condition as MTT, trypan blue, proliferation, and apoptosis). But we didn’t detect some significant change. We thought the reason was that we gave enough time to cells recovery. Therefore, we further directly detect the change of intracellular ROS and DNA damage immediately after the irradiation of blue light. So these results means the ROS generation and DNA damage occurred immediately after the irradiation, which was not an irreversible effect. This result is our prior experimental result is not presented in this paper, so we changed this sentence to " there was no significant change after three days irradiation in our previous study " please refer to page 17 line 350.

5. In the abstract, authors state “low energy density blue light multiple exposure is still harmful to our cells even though the effect is lower than short wavelength high energy blue light.” Please cite evidence of this and discuss in the discussion.

Reply: Thank you very much for your comment. We think your comments are very helpful. We have mentioned the harmful effects of short wavelengths on cells from previous studies in our discussion, but we did not specifically compare the difference between long and short wavelengths. We thought that short wavelength led to apoptosis and longer wavelength didn’t induce apoptosis, is the relatively lower harmful effect. However, this idea is indeed arbitrary. So that we removed our arbitrary description in abstract. “Together, these results indicate that longer wavelength and low energy density blue light multiple exposure is still harmful to our cells.” please refer to page 2 line 26 . 

Figures

1. Please define all abbreviations used in figures in the key (ex/ NL, B1h, B2h)

Reply: Thank you very much for your comment. We added the abbreviation in the legends of figures. 

“NL: non-light treatment; BL: blue light irradiation; B1h, B2h: blue light irradiation 1, 2 hours.” in figure 1, 

“NL: non-light treatment, BL: blue light irradiation.” in figure 2, 

“NL: non-light treatment, BL: blue light irradiation.” in figure 3, 

“NL: non-light treatment, BL: blue light irradiation, tyr: tyrosinase.” in figure 4.

---

## [Editor Report · Decision Letter 2]

17 Jan 2023

Low Energy Multiple Blue light-emitting diode light Irradiation promotes melanin synthesis and induces DNA damage in B16F10 melanoma cells

PONE-D-22-27769R2

Dear Dr. Sakamoto,

We’re pleased to inform you that your manuscript has been judged scientifically suitable for publication and will be formally accepted for publication once it meets all outstanding technical requirements.

Kind regards,

Yi Cao

Academic Editor

PLOS ONE
---

## [Editor Report · Acceptance letter]

23 Jan 2023

PONE-D-22-27769R2 

Low Energy Multiple Blue light-emitting diode light Irradiation promotes melanin synthesis and induces DNA damage in B16F10 melanoma cells 

Dear Dr. Sakamoto:

I'm pleased to inform you that your manuscript has been deemed suitable for publication in PLOS ONE. Congratulations! Your manuscript is now with our production department. 

Kind regards, 

on behalf of

Dr. Yi Cao 

Academic Editor

PLOS ONE